# Flavanols from Nature: A Phytochemistry and Biological Activity Review

**DOI:** 10.3390/molecules27030719

**Published:** 2022-01-22

**Authors:** Yu Luo, Yuqing Jian, Yingkai Liu, Sai Jiang, Daniyal Muhammad, Wei Wang

**Affiliations:** TCM and Ethnomedicine Innovation & Development International Laboratory, Innovative Materia Medica Research Institute, School of Pharmacy, Hunan University of Chinese Medicine, Changsha 410208, China; luoyu1998111@163.com (Y.L.); yingkailiu@hotmail.com (Y.L.); saijiang626@hotmail.com (S.J.); Daniyaldani151@yahoo.com (D.M.)

**Keywords:** flavanols, flavonoids, natural products, biological activities

## Abstract

Flavanols, a common class of secondary plant metabolites, exhibit several beneficial health properties by acting as antioxidant, anticarcinogen, cardioprotective, anti-microbial, anti-viral, and neuroprotective agents. Furthermore, some flavanols are considered functional ingredients in dairy products. Based on their structural features and health-promoting functions, flavanols have gained the attention of pharmacologists and botanists worldwide. This review collects and summarizes 121 flavanols comprising four categories: flavan-3-ols, flavan-4-ols, isoflavan-4-ols, and flavan-3,4-ols. The research of the various structural features and pharmacological activities of flavanols and their derivatives aims to lay the groundwork for subsequent research and expect to provide mentality and inspiration for the research. The current study provides a starting point for further research and development.

## 1. Introduction

Various flavanols have been isolated from Nature over the past few decades. As their potential beneficial properties have been recognized, they have been increasingly studied by scientists worldwide [1]. Up to now, flavanols have been found in common foods, including cereals, legumes, fruits, vegetables, forages, hops, beers, red wine, tea, cocoa, grapes, and apples [2]. In addition, flavanols were used as quality markers for nuts and cereals as well. Therefore, there is a growing awareness of the benefits of natural flavanols. As a distinct sub-group of flavonoids, flavanols are broadly characterized by the absence of a no double bond between C-2 and C-3 and the absence of a carbonyl group on the C ring (C-4), while featuring a hydroxyl group(s) on C-3 or C-4. Given the above structural characteristics, four main types of flavanols have been found in Nature: (i) flavan-3-ols, (ii) flavan-4-ols, (iii) isoflavan-4-ols, and (iv) flavan-3,4-ols, whose basic skeletons are shown in Figure 1. Flavan-3-ols are the most commonly reported among the four types, followed by flavan-4-ols and flavan-3,4-ols, and lastly, isoflavan-4-ols. The classification and quantity of each category of flavanols are displayed in Figure 2. Due to aggregation, oligomerized flavan-3-ols and oligomerized flavan-3,4-ols have also been discovered in succession. The four subtypes of flavan-4-ols are based on the position of the linked saccharide or the seven-membered oxygen heterocycle between the sugar and aglycone. In general, 121 flavanols and their derivatives were found from the natural plant kingdom, distributed in 52 species and 29 families (Figure 2). They range from simple monomers to oligomers, and from aglycones to glycosides. Through the analysis and data statistics, plants rich-in flavanols are mainly include 10 species (*Pronephrium*
*penangianum*, *Camellia sinensis*, *Theobroma cacao*, *Astragalus membranaceus*, *Rhoicissus tridentate*, *Juniperus communis* var. *depressa*, *Camellia sinensis* var*. assamica*, *Livistona chinensis**, Thespesia populnea* and *Wisteria floribunda*), are marked in blue color (Figure 2) and should be considered in the future development of natural flavanols. Several of these plants were used in the treatment of various diseases. For example, *Pronephrium*
*penangiana* (previously named *Abacopteris*
*penangiana*) is widely distributed in the south of China and used as a folk medicine to treat blood circulation stasis, barriers to the study of blood flow, edema and inflammation for patients suffering from metabolic syndrome [3]. *Astragalus mongholicus* (also named Huangqi in China), a well-known herb amongst traditional Chinese medicine practitioners, has been used to improve resistance to infections, in immunological disorders and viral infections, and used as a hepatoprotective heart tonic, and in the treatment of nephritis, and diabetes [4]. *Hovenia dulcis* is commonly used in Korean and Chinese traditional medicine to treat liver diseases and as a detoxifying agent for alcohol poisoning. *Garcinia dulcis* is an Asian medicinal plant used in folk medicines and its stem bark has been used as an anti-inflammatory agent in Thailand [5].

Although flavanols feature good bioactivity, there is no systematic and comprehensive review for phytochemistry and biological activities [6,7]. This review is focused on the phytochemical investigations and biological activities of naturally occurring flavanols, covering the literature from 1974–2021.

## 2. Flavan-3-ols

Flavan-3-ols can be classified into simple flavan-3-ols, alkaloid flavan-3-ols and oligomeric flavan-3-ols by analyzing the structural characteristics of published compounds (Table 1, Figure 3).

### 2.1. Simple Flavan-3-ols

As the most common group of flavanols in the diet, simple flavan-3-ols are found in a wide range of raw-food materials [45]. Due to their significant health benefits, these ingredients are candidates in developing dietary recommendations [46]. Representing the most common flavonoid consumed in the American diet, the flavan-3-ols and their condensation products are regarded as functional ingredients in various beverages, herbal remedies and supplements [47]. They often occur in food and contribute to astringency, bitterness, sourness, sweetness, salivary viscosity, aroma, and color formation as food quality parameters. Furthermore, they are also used as food quality parameters. Moreover, flavan-3-ols also improve food’s microbial, oxidative, and heat stability [47].

Structurally, flavan-3-ols rarely occur as glycosides, they are characterized by an absence of a double bond between C-2 and C-3, the presence of a hydroxyl group on position C-3, and the absence of the C-4 carbonyl in ring-C. Thus, flavan-3-ols have two chiral centers at C-2 and C-3.

At present, a total 34 simple flavan-3-ols have been reported. Generally speaking, based on the two chiral centers C-2 and C-3 of ring-C, flavan-3-ols can exist as four possible stereoisomers: (*2R*,*3S*), (*2S*,*3R*), (*2R*,*3R*), (*2S*,*3S*), as shown in Figure 3. Most of the natural flavan-3-ols are *2R* isomers (compounds **1**–**11****, 17**–**32**) while *2S* isomers (**12**–**16, 33, 34**) only occur in a few plants such as *Euonymus laxiflorus, Ulmus*
*davidiana* and *Plicosepalus*
*curvifloru*. Hydroxyl is the most common group on ring-A and ring-B of flavanols. The substitution positions of hydroxyl groups on the ring-A of most simple flavan-3-ols are mainly at C-5 and C-7, while a few compounds like **9**, **10** and **17** only have one hydroxyl group at the C-7 position. The main substitution positions of hydroxyls on ring-B are C-3′, C-4′ and C-5′. Furthermore, hydroxyl groups at different positions could also be methylated, esterified with gallic acid, and generate glycosidic bonds with different monosaccharides such as glucose, rhamnose and allose.

### 2.2. Alkaloid Flavan-3-ols

In the review, a group of exceptional flavan-3-ols attached to pyrrolidone, indole moiety and skytanthine moieties (compounds **35**–**47**) is uniformly classified as alkaloid flavan-3-ols [48]. They are usually linked with five-membered lactam rings at the C-6 or C-8 positions of ring-A. Interestingly, a new chiral center is formed naturally based on the γ-position carbon of the lactam ring attached to the aromatic ring. These alkaloid flavan-3-ols often appear in the form of a couple of epimers with extremely similar NMR data. In addition, an ethyl group was usually connected to the nitrogen atom and galloyl group as a common moiety link to the C-3 position. Compound **44,** also named kopsirachin, featured two skytanthine moieties linked to the C-6 and C-8 positions, respectively. Compound **45** is characterized by a monoterpene indole unit connected at the C-8 position, and compound **46** features an acylamino indole unit connected at the C-4 position. They share a common biosynthetic precursor, tryptophan. Compound **47** is a unique dimer in alkaloid flavan-3-ols, having an *N*-ethyl-2-pyrrolidinone moiety linked to the C-8 position, and it also has a dimeric structure.

### 2.3. Oligomeric Flavan-3-ols

Some basic flavan-3-ols can be condensed to form oligomeric flavan-3-ols, usually found in cereals, legumes, fruits, vegetables, forages, and beverages. They play a crucial role in the generation of phenyl-γ-valerolactones and phenylvaleric acids, which feature appealing health benefits.

So far, several oligomeric flavan-3-ols, including dimers, trimers, tetramers, pentamers and hexamers, have been reported. Depending on the inter-flavanol linkages, six types of dimers can be distinguished. Compounds **48**–**56** are formed by condensation between the C-4 of the upper monomer and the C-8 or C-6 of the adjacent lower units. Additionally, in compounds **48**–**51**, two pairs of isomers are observed, possessing an additional C-2–*O*–C-7 linkage between monomeric ones. Compounds **59** and **61** were determined to be dimeric flavan-3-ols through a methylene linkage between C-8 and C-8 or C-6 and C-6 of two basic units. Compound **60** is determined to be a dimeric flavan-3-ol glycoside. The dimeric flavan-3-ols **62**–**65** with the characteristic 5/7 or 7/5 fused ring benzotropolone skeleton system, namely theaflavins, were generated by a special pattern between ring-C of two basic flavan-3-ols. It has been reported that theaflavins heavily influence the astringency of black teas and therefore affect tea tasting evaluations. In theaflavins, the most common group linked at 3-OH is galloyl. Two types of trimers can be classified by inter flavanol linkage inter flavan-3-ols. Compounds **66**–**69** consist of three flavan-3-ol units linked by a C-4–C-8 bond between basic flavan-3-ols. In contrast, compounds **70**–**72** possess an additional C-2–*O*–C-7 linkage between two simple units. Compound **73** is the only tetrameric flavan-3-ol isolated from *Theobroma cacao*, in which four (−)-epicatechin units are linked by C-4–C-8 bonds. Pentameric and hexameric flavan-3-ols are mainly linear oligomers formed via C-4–C-8 bonds and occasionally contain a branched unit via a C-4–C-6 bond between flavanols.

## 3. Flavan-4-ols

As the name implies, flavan-4-ols have a hydroxyl group at the C-4 position of ring C (in Table 2, Figure 4). Only a few of these compounds, such as **77** and **78,** exist in the form of aglycones. Other flavan-4-ols **79**–**100** with the same 6,8-dimethyl features usually possess glucose or glucuronic acid moieties attached to C-5 or C-7 of ring-A to form phenolic *O*-glycosides. Notably, some 5-*O*-glucosides or 7-*O*-glucosides of flavan-4-ols can generate seven-membered oxygen heterocycles by dehydration between the 2-OH of glucose and an aglycone hydroxyl.

Until now, a total of 24 flavan-4-ols have been described in the literature, most of them were isolated from *Pronephrium penangianum*, which is a traditional national herb medicine of the Tujia nationality in Hunan Province of China and has a good pharmacological activity on relaxing muscles and tend on stop romoting blood circulation, stanching bleeding and relieving pain in the Tujia system of medicine. Because of its red chicken blood color, the Tujia people name it “jixueqi”. With the further investigation of “jixueqi”, botanists found that its Chinese name is “pizhenxinyuejue”.

## 4. Isoflavan-4-ols

Isoflavan-4-ols (compounds **101**–**109**) are a small group of flavanols isolated from the *Astragalus* genus (in Table 3, Figure 5), which is the largest genus in the Leguminosae family. Inconsistent with flavan-4-ols, the ring-B of isoflavan-4-ols is directly linked to C-3, and the 4-OH does not exist in the form of free hydroxyl. Consequently, a furan ring is easily formed between C-3, C-4 of ring-C and C-1′, C-2′ of ring B.

## 5. Flavan-3,4-ols

Flavan-3,4-ols are classified into simple flavan-3,4-ols and dimeric flavan-3,4-ols based on aggregation (in Table 4, Figure 6).

### 5.1. Simple Flavan-3,4-ols

Only a small percentage of flavan-3,4-ols have been found in Nature. They present a hydroxyl group or a glycoside on position C-3 and C-4 and generate three chiral centers at C-2, C-3 and C-4. Based on the chiral centers, four compounds (110–113) are distinguished by their absolute configuration. However, compounds **114** and **115** have a six-membered oxygen heterocyclic ring linking C-2, C-3, C-1′ and C-2′.

### 5.2. Oligomeric Flavan-3,4-ols

Oligomeric flavan-3,4-ols consists of several simple units and have additional chiral centers at C-3 and C-4 of all terminal monomer. Compounds **118**–**121** differ from other flavonoids by the presence of a fourth ring formed by a one-carbon bridge between an oxy functional group at C-3 of a flavonoid and a carbon of the flavonoid B ring. All the oligomeric flavan-3,4-ols have been isolated from *Acacia peuce*, *A. carneorum* and *A. crombiei*.

## 6. Pharmacological Activities

The pharmacological and biological properties of flavanols have been extensively studied to reveal their anti-oxidation, anti-inflammatory, anti-cancer, anti-viral effects, and protective cardiovascular properties, among others.

### 6.1. Anti-Oxidation

Flavanols are unanimously considered a natural source of antioxidant compounds. Flavan-3-ols exert their antioxidant effects via three main mechanisms. Flavan-3-ols have been shown three main mechanisms of antioxidants: (i) free radical scavenging, (ii) chelation of transition metals, (iii) inhibition of enzymes [68]. Free radicals are highly reactive species that contain an unpaired electron in the valence shell. The activity is a redox transition involving the donation of a single electron (or H-atom, equivalent to donation of an electron and an H^+^) to a free radical species(R^•^). The scavenging of the hydroxyl radical (OH^•^), superoxide (O_2_^•−^), and H_2_O_2_ is a common mechanism for anti-oxidation. DPPH (1,1-diphenyl-2-picrylhydrazyl) and ABTS [2,2′-amino-di(2-ethyl-benzothiazoline sulphonic acid-6)ammonium salt] are the most frequently encountered stable free radicals. Their antioxidative power and activity are measured by their ability to deactivate free radicals [69]. Free state iron and copper are usually combined with free radicals through Fenton and Haber-Weiss reactions. Flavan-3-ols bind to such divalent transition metals and effectively reduce the concentration of these cations, attenuating the extent of their oxidative activity [6]. Enzyme inhibition is also a vital mechanism to flavan-3-ols, especially for oligomeric flavan-3-ols [70,71,72]. They exhibit antioxidant activity through inhibition of prooxidative enzymes and lipoxygenase. (-)-epicatechin played a significant role in cellular redox regulation. It has been found to modulate the expression of enzymes producing superoxide anion, hydrogen peroxide, and nitric oxide [73], together with NADPH oxidases [74,75], xanthine oxidase [75], cyclooxygenase [76], and nitric oxide synthases (NOS). Furthermore, (−)-epicatechin interacts with cell membranes [77,78,79], reducing the lipid oxidation rate and regulating membrane-bound enzyme/receptor activity [80]. It also improves mitochondrial function and structure, decreasing the formation of superoxide anion and hydrogen peroxide. Moreover, it inhibits endoplasmic reticulum (ER) stress [81], leading to increased oxidant production. In addition, (−)-epicatechin upregulates antioxidant defense mechanisms via Nrf2 (NF-E2-related factor 2) activation [82,83,84]. Epicatechin, epigallocatechin, epicatechin gallate, and epigallocatechin gallate demonstrate excellent free radical scavenging activity. (+)-Catechin, (−)-catechin, (−)-afzelechin and (−)-epicatechin showed potent antioxidant activity, measured by the concentration of the extract required for 50% reduction in DPPH radical absorbance (13.5, 13.6, 21.8 and 20.9 μM, respectively). In contrast, the positive control Trolox required 48.8 μM [20]. Compounds **18**, **50**, **51**, **57**, **58**, **66**, **70**, **71**, **73** exhibited anti-oxidant activities attributed to their inhibitory effects on nicotinamide adenine dinucleotide phosphate dependent lipid peroxidation in microsomes and on the autoxidation of linoleic acid. These effects are related to the radical-scavenging activity in the peroxidation chain reactions [46]. Compounds **18**, **50**, **51**, **57**, **58**, **66**, **70**, **71**, **73** have inhibitory effects of on the autoxidation of linoleic acid initiated by the addition of V-70 with an IC_50_ value of 0.62–5.3 μg / mL**.** These compounds have radical-scavenging effects on the DPPH radical with an EC_50_ value of 1.4–3.9μM. Compound **18** demonstrated dose-dependent scavenging potential and have particularly advantageous capacity against peroxynitrite (3.37–13.26 mmol AA/g), hydroxyl radical (5.03–8.91 mmol AA/g) and superoxide radical (3.50–5.50 mmol AA/g) (ascorbic acid (AA) was used as positive control.) [85]. Many flavan-4-ol glycosides also showed remarkable antioxidant and free radical scavenging properties. D-Gal is a normal, reducing sugar in vivo which can be oxidized into free radicals when present in high levels, causing oxidative stress and cellular damage. Abacopterin E, one of the flavan-4-ol isolated from *Pronephrium*
*penangiana*, can scavenge free radicals, modulate the activities of antioxidant enzymes and reduce the lipid peroxidation in the hippocampus of D-Gal-treated mice [57].

### 6.2. Anti-Inflammation

To date, several experiments have demonstrated the anti-inflammatory activity of flavanols. NO levels, nuclear factor kappa-B (NF-κB) and tumor necrosis factor-alpha (TGF-α) are well-established markers for the inflammatory process [86]. After infection, NO and several interleukins are generated by macrophages to combat the infectious agent. NF-κB regulates the expression of the cytokines, including interleukin-1 beta (IL-1β) and TNF-α, which are essential mediators of chronic inflammation and are implicated in leukocytosis, hyperplasia, and tissue break down [12]. The nuclear transcription factor NF-κB signaling pathway is the main feature of the inflammatory responses in oxidative-dependent diabetic aortic pathology. It is a major proinflammatory switch that can regulate the efflux of cytokines, such as monocyte chemoattractant protein (MCP)-1 and vascular endothelial growth factor (VEGF). (+)-Catechin inhibits the release of NO by two pathways: direct elimination of NO activity and inhibition of nitric oxide synthase (NO-iNOS) protein expression [87]. In the studies performed by Bui HuuTaia and Trinh Nam Trung, (+)-catechins’ anti-inflammation activities were also reported [12]. (+)-Catechin and (−)-epicatechin have inhibitory effects on TNF-α-induced NF-κB activation in HepG2 cells with IC_50_ values of 14.1, 16.5 μM, respectively. The study also showed that (+)-catechin, (+)-afzelechin, (−)-epiafzelechin and dulcisflavan inhibit the pro-inflammatory inducible nitric oxide synthase and cyclooxygenase-2 proteins in TNF-α-stimulated HepG2 cells at concentrations as low as 0.1 μM. Catechin-5-*O*-β-d-glucoside showed anti-inflammatory activity with an inhibition rate of over 33.0% ± 4.0 [23]. Theaflavins exhibit anti-inflammation effects by down-regulating the activation of NF-κB in macrophages. Theaflavin-3,3′-digallate, which contains two gallic acid moieties, exhibited the strongest anti-inflammatory activity as judged by its suppression on inducible NO synthase induction. Theaflavin, which featuring no gallic acid moiety, exhibited the least inhibitory effects [88]. From these results, we conclude that the gallic acid moiety is crucial for the anti-inflammatory activity of theaflavins. The same mechanism is observed in epigallocatechin gallate. It has been reported that (+)-catechin, (−)-epicatechin and its dimeric fraction inhibit NF-κB by reducing the production of IL-2 in T cells during pro-inflammatory stimulation [89,90]. (−)-Gallocatechin-3-gallate, (−)-gallocatechin and gallic acid inhibit lipopolysaccharide (LPS)-activated induction of nitric oxide synthase (NOS) in mouse peritoneal macrophages [91], because their gallocatechin and hydroxyl groups have anti-inflammatory effects by inhibiting NF-κB activation through NOS transcription [88]. Furthermore, 6-(2-pyrrolidinone-5-yl)-(−)-epicatechin and 8-(2-pyrrolidinone-5-yl)-(−)-epicatechin have inhibitory activities against AGEs formation, with IC_50_ values of 13.5, 17.9 μg / mL. A few flavan-4-ols also demonstrate significant anti-inflammation activities. Abacopterin A exerts its anti-inflammatory properties by inhibiting NF-κB expression and reducing the inflammatory response. These findings represent a potential agent of APA (Abacopterin A) for the treatment of inflammation [92].

### 6.3. Anti-Cancer

Cancer is one of the most serious diseases in the world and is closely related to improper regulation of apoptosis. Apoptosis is an active genetically determined process that automatically ends cell life, and is often called programmed cell death. A damaged or blocked apoptotic pathway results in uncontrolled cell division eventually leading to tumor formation [6]. Epigallocatechin-3-gallate exhibited strong effects on the induction of apoptosis and regulation of cell cycle in human and mouse carcinoma cells. Epigallocatechin-3-gallate exhibits strong apoptotic induction effects and regulates the cell cycle in human and mouse carcinoma cells. Epigallocatechin-3-gallate, epigallocatechin-3-gallate, epigallocatechin, and epicatechin-3-gallate, also led to the formation of inter nucleosomal DNA fragments and resulted in apoptosis in A431 cells and SMMC-7721 cells [93,94]. (2*S*,3*S*)-3,5,7,3′,5′-pentahydroxyflavane has significant antiproliferative effects against HL-60 and CNE-1 with IC_50_ of 0.2 ± 0.01 and 1.0 ± 0.1 μM [16]. 12-*O*-tetradenoylphorbol 13-acetate (TPA) is a potent tumor promoter. Oligomeric flavanols can inhibit TPA-induced ornithine decarboxylase (ODC) activity to against cancer. Furthermore, this inhibition increases with the degree of oligomeric flavan-3-ols (trimer > dimer > monomer) [95]. Abacopterin C, abacopterin D, eruberin C and triphyllin A have exhibited a strong anticancer activity in vitro with a more than 60% inhibition rate of HepG2 at a dose of 50 μM. Based on the strong anti-cancer properties of abacopterin C, our team showed a new drug-delivery system using a compound derived from *Pronephrium penangianum* (abacopterin C) for the treatment of cervical cancer. The system provides a potential synergistic platform for cancer therapy. An enhanced silver nanocluster system for cytochrome c detection and natural drug screening was also established. The release of Cyt c from apoptotic tumor cells was monitored by the novel fluorescence-enhanced nanoprobe, DNA-AgNCs@tween 80, confirming its effectiveness. Abacopterin C is a potential drug for treating anti-colon tumors [96].

### 6.4. Anti-Viral

Epigallocatechin-3-gallate and theaflavins (theaflavin, theaflavin-3-gallate, theaflavin 3′-gallate, and theaflavin-3,3′-digallate) exhibit antiviral properties against various viruses. Epigallocatechin-3-gallate effectively inhibits viral infections such as PRRSV (porcine reproductive and respiratory syndrome virus) [97], hepatitis C virus (HCV) [98], (west nile virus) WNV, (human immunodeficiency virus) HIV [99] and so on. Theaflavins showed inhibitory effects against HCV [100], TMV (Tobacco Mosaic Virus) [101] and other viral [102] infections.

COVID-19 is a viral disease that affects the epithelial cells of the respiratory system and causes inflammation of the mucosal membrane. Its outbreak poses a threat to the people’s lives all over the world. COVID-19 is an infectious disease caused by a novel positive-sense single-27 stranded RNA corona virus called SARS-CoV-2. 3CL pro, a vital enzyme found in SARS-CoV-2, is a druggable target owing to its important function in viral cell replication. (−)-Epicatechin-3-*O*-gallate and theaflavins inhibit 3CL pro at lesser concentrations and also exhibit good binding to this target [103]. In molecular docking, Epigallocatechin-3-gallate and theaflavin-3-gallate showed that the tea polyphenols had good docking scores with the possible targets of COVID-19 [104]. (−)-Epicatechin-3-*O*-gallate and theaflavins may be potential 3CL pro inhibitors for the treatment of COVID-19.

### 6.5. Cardiovascular Protection

Cardiovascular and cerebrovascular diseases are major life-threatening conditions, and lipid-lowering therapy has been widely used to reduce the incidence of these diseases. ACE activity is tightly linked to cardiovascular and cerebrovascular diseases. The inhibitory activity of angiotensin I-converting enzyme (ACE) was examined with metabolites of compound **21** produced by intestinal bacteria, together with tea catechins [105]. Oligomeric flavan-3-ols (dimer and hexamer) and epigallocatechin significantly inhibited enzyme activity and ACE activity [106]. Another molecular cardioprotective mechanism of oligomeric flavan-3-ols is LDL oxidation inhibition (low-density lipoprotein). LDL in the blood may accumulate within the arterial wall and is prone to oxidation [107]. Flavan-3-ols can reduce cholesterol levels, restore lipid balance, and prevent the conversion of LDL to harmful oxidized chemical states [108].

### 6.6. Vascular Protection

Flavan-3-ols have therapeutic benefits in neurodegenerative conditions such as Alzheimer’s and Parkinson’s diseases. Alzheimers disease (AD) is a type of dementia, is one of the most common neurodegenerative diseases [109]. It is mostly characterized by extensive protein aggregation with amyloid-beta deposition in senile plaques and tau aggregation in neurofibrillary tangles. The ubiquitin-proteasome system (UPS) and theautophagy-lysosomal pathway are the two major catabolic pathways in eukaryotic cells. The UPS controls the degradation of cytosolic and nuclear proteins, including short-lived proteins, whereas autophagy is responsible for the clearance of protein aggregates and damaged organelles. Due to an age-dependent decline in their activity, aberrant proteins accumulate, contributing to the onset and development of age-related disorders, such as AD. Phenyl-γ-valerolactonesare microbiota-derived metabolites of flavan-3-ols; some of them activate the autophagic pathway in the SH-SY5Y control cell line. According to the cell line, we can find the change of LC3II (the lipidated form of LC3 that localizes in autophagosomal membranes) and p62 (a substrate of autophagy that accumulates in cells when autophagy is inhibited) [110]. SH-SY5Y control cells showed increased levels of LC3II but almost no change in the expression of the p62 protein, suggesting the activation of the early steps of the autophagic pathway in this cell line [111].

### 6.7. Miscellaneous Activities

Flavanols have also been associated with other beneficial health effects by acting as anti-microbials, anti-virals, anti-parasitics, anti-AChE and anti-alkaline phosphatase. Chunget al. summarized the anti-microbial effects of several tannin extracts on yeast, filamentous fungi, bacteria, and viruses [112]. (−)-8-*N*-ethyl-2-pyrrolidinone-epiafzelechin-3-*O*-gallate showed inhibitory activity against AChE with IC_50_ values of 14.23 μM. (−)-epicatechin-3-*O*-β-d-allopyranoside has alkaline phosphatase (ALP) activity by increasing mineralization up 120.8%, and 134.6% at 80 and 100 µM.

### 6.8. Metabolism and Pharmacokinetics

Phenyl-γ-valerolactones (PVLs) and their related phenylvaleric acids (PVAs) are the main metabolites of flavan-3-ols. In 1958, PVLs were first reported to be flavan-3-ol metabolites by Oshima and Watanabe [113]. As more PVLs and PVAs were discovered, their activities in humans have attracted attention. 5-(3′, 5′-dihydroxyphenyl)-γ-valerolactone, the main ring-fission metabolite of compound **21**, showed slightly higher blood-brain barrier (BBB) permeability than **21** and had neuritogenic activity in SH-SY5Y cells at 0.05 µM [114]. In addition, metabolites of **21** degraded by rat intestinal flora have been confirmed to have antioxidative activity [115]. 5-(3′,4′-dihydroxyphenyl)-γ-valerolactone is the major microbial metabolite of proanthocyanidins. By down regulating TNF-α-stimulated expressions of the two biomarkers of atherosclerosis vascular cell adhesion molecule-1 and monocyte chemotactic protein-1, the metabolite prevents THP-1 monocyte-endothelial cell adhesion, activation of nuclear factor kappa B transcription and phosphorylation of I kappa-B kinase and IκBα [116]. The radical scavenging abilities of the metabolites were found to be stronger than or nearly equal to those of Trolox (positive medicine) [114].

## 7. Conclusions

The review summarized 121 flavanols among flavan-3-ols, flavan-4-ols, isoflavan-4-ols and flavan-3,4-ols. Few systematic studies have reported flavanols among four types, which highly impact biological activities. The Theaceae plants *Camellia sinensis* and the Thelypteridaceae plant *Pronephrium penangianum* possess the most abundant flavanols. *Pronephrium penangianum* harbors flavan-4-ols as its major constituents. *Camellia sinensis* var. *assamica*, another Theaceae plant, is the main source of alkaloid flavan-3-ols. Most of them showed strong inhibitory activity on α-glucosidase and acetyl-cholinesterase. *Astragalus* root is a very old and well-known drug in traditional Chinese medicine. The *Astragalus* genus has been reported to be rich in flavanols as one of its characteristic constituents. Another notable species is *Acacia sensustricto*. It is a dominant species in most Australian landscapes, with reported uses in the traditional medicine of Australian Indigenous people, and in traditional medicine systems of the Pacific Islands and Asia. Five plants consisting of *Acacia melanoxylon, Acacia peuce, Acacia carneorum, Acacia crombiei, Acacia fasciculifera* contain flavanols. These findings may be of important chemotaxonomic significance and provide a foundation to find more flavanols.

Most flavanols exhibit anti-oxidative, anti-inflammatory, anti-cancer, anti-viral activities, and have vascular protective effects on the cardiovascular system. Catechins and their derivatives generally exhibit strong antioxidant activity based on their structures, bioactivities, and pharmacokinetics. The most promising flavanols for medicinal development include the anti-cancer agents abacopterin A and abacopterin C. With the rapid development of nanomaterials chemistry, there are abundant types of nanoformulations, such as nanoflabers, nanoliposomes, and micelles, that can be used to improve flavanols targeting and druggability. (−)-epicatechin-3-*O*-gallate and theaflavins were previously reviewed for their antiviral activities, they are potential antiviral agents which should be explored as treatment and prophylactic alternatives for COVID-19. The potential of the flavanols for the treatment of cardiovascular, cerebrovascular diseases, Alzheimer’s and Parkinson’s diseases are also noticeable. Furthermore, the data of the genome of all kinds of flavanols, in parallel with the use of transcriptomic, proteomic, and metabolomic technologies, will aid in the elucidation of the genes involved in pharmacology activities.

However, the study on flavanols are far from enough. On the one hand, the content and purity of the active compounds isolated by traditional methods are usually low and subject to varying environmental factors, making it difficult to meet the huge demand of the international market. This highlights the need to develop new strategies for the preparation of these characteristic compounds. On the other hand, the biggest obstacle to large-scale production of flavanols is the lack of knowledge regarding the biosynthetic pathways. Meanwhile, a deeper work of isolation and biosynthetic pathways should be explored, and more pharmacological activities studies are necessary to define the active constituents of flavanols. The discovery of new compounds and transcriptomic, proteomic, and metabolomic technologies of flavanols are important and promising, such researches can provide clues for the development of new drugs. In this paper, the current research status of flavanols was reviewed in order to provide some scientific theoretical basis and reference for the follow-up research of flavanols. So far, few results have been published in this regard. It is expected that in the next few years more research will be carried out on this topic.

## Figures and Tables

**Figure 1 molecules-27-00719-f001:**
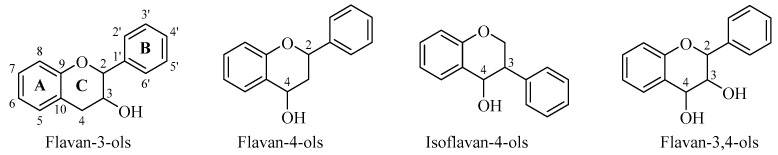
Basic skeletons of flavanols.

**Figure 2 molecules-27-00719-f002:**
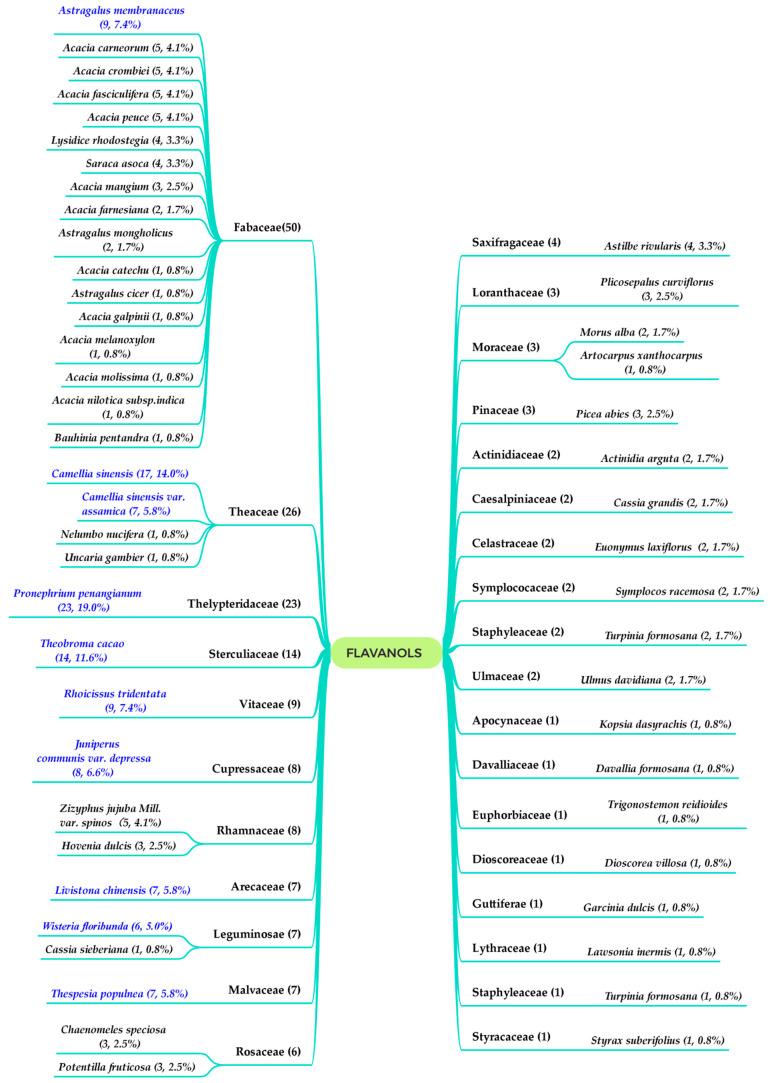
Plant sources of 121 flavanols (Families and species and the corresponding quantity of compounds).

**Figure 3 molecules-27-00719-f003:**
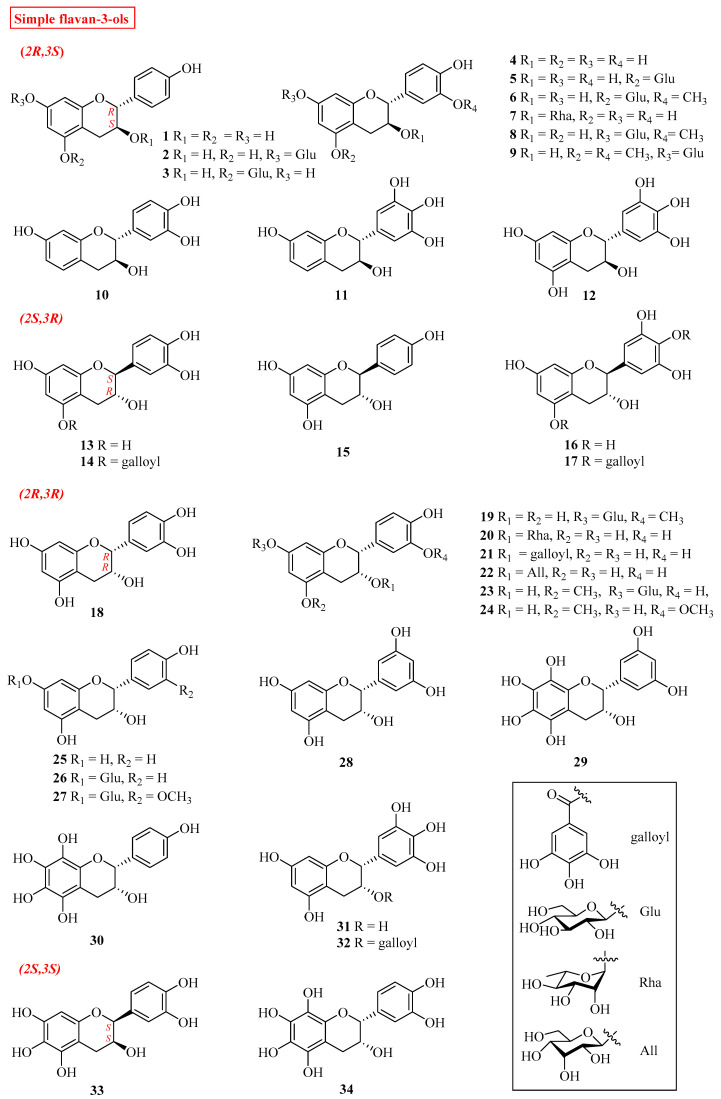
Structures of flavan-3-ols.

**Figure 4 molecules-27-00719-f004:**
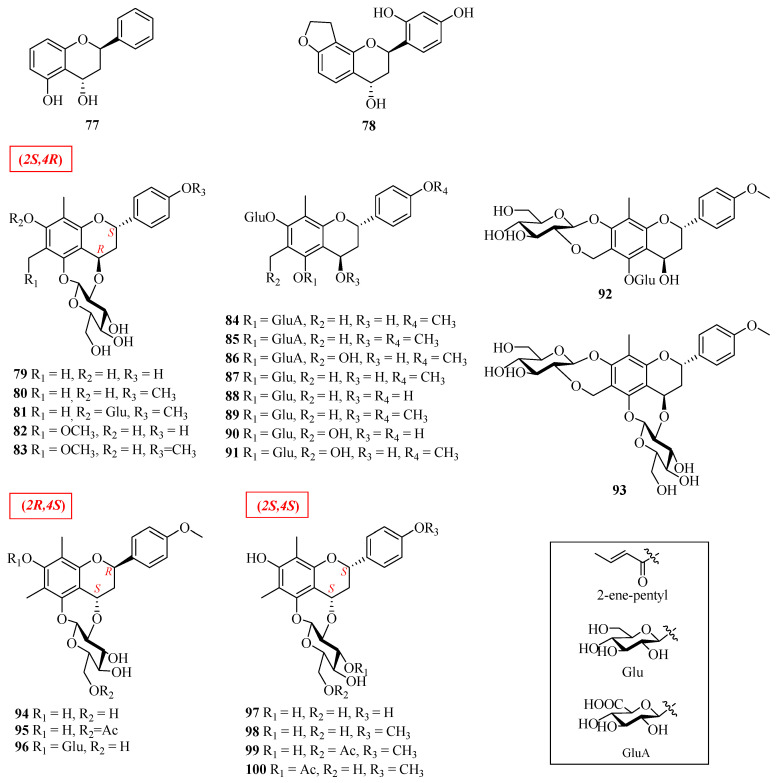
Structures of flavan-4-ols.

**Figure 5 molecules-27-00719-f005:**
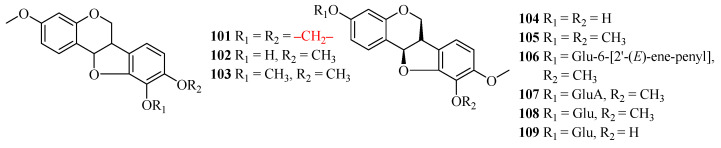
Structures of isoflavan-4-ols.

**Figure 6 molecules-27-00719-f006:**
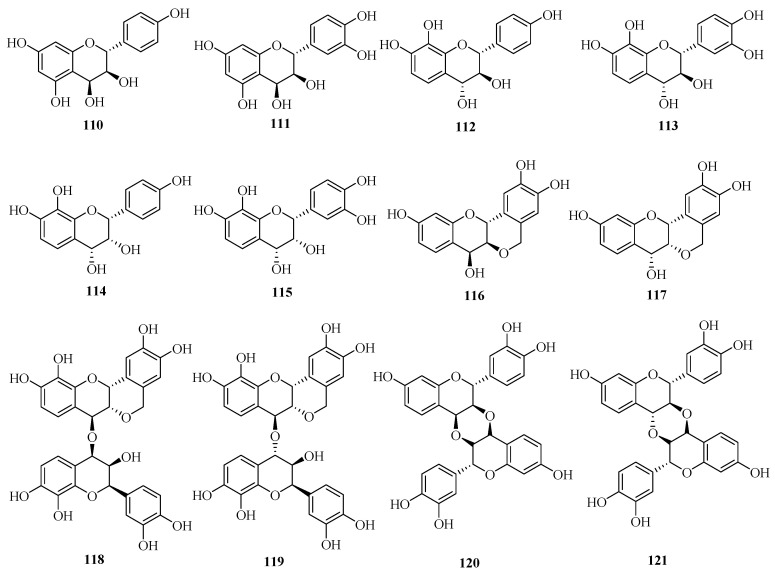
Structures of flavan-3,4-ols.

**Table 1 molecules-27-00719-t001:** The numbers, plant sources, corresponding organs of flavan-3-ols.

No.	Name	Plant Source	Plant Organ	Ref.
**1**	(+)-afzelechin	*Wisteria floribunda*	fruit peels	[8]
		*Nelumbo nucifera*;	seed epicarp	[9]
		*Hovenia dulcis*;	stem barks	[10]
		*Zizyphus jujuba Mill.*	roots	[11]
		*var. spinosa*		
		*Livistona chinensis*	fruits	[12]
		*Thespesia populnea*	barks	[13]
**2**	(+)-afzelechin-7-*O*-β-d-glucopyranoside	*Juniperus*	stems and leaves rhizomes	[14]
		*communis var. depressa*		
		*Dioscorea villosa*		[15]
**3**	(+)-afzelechin-5-*O*-β-d-glucopyranoside	*Wisteria floribunda*	fruit peels	[8]
		*Dioscorea villosa*	rhizomes	[15]
**4**	(+)-catechin	*Wisteria floribunda*	fruit peels	[8]
		*Astilbe rivularis*	rhizomes	[16]
		*Uncaria gambier*	leaves	[9]
		*Rhoicissus tridentata*	roots	[17]
		*Zizyphus jujuba Mill.*	roots	[11]
		*var. spinosa*		
		*Euonymus laxiflorus*	trunk barks	[18]
		*Livistona chinensis*	fruits	[12]
		*Thespesia populnea*	barks	[13]
		*Chaenomeles speciosa*	fruits	[19]
		*Picea abies*	root barks	[20]
		*Lawsonia inermis*	aerial parts	[21]
**5**	catechin-5-*O*-β-d-glucoside	*Chaenomeles speciosa*	fruits	[19]
**6**	(+)-3′-*O*-methylcatechin-5-*O*-β-d-glucopyranoside	*Juniperus*	stems and leaves	[14]
		*communis var. depressa*		
**7**	3′- deoxycatechin-3-*O*-α-l-rhamnopyranoside	*Saraca asoca*	barks	[22]
**8**	(+)-4′-*O*-methylcatechin 5-*O*-β-d-glucopyranoside	*Juniperus*	stems and leaves	[14]
		*communis var. depressa*		
**9**	(+)-gallocatechin	*Rhoicissus tridentata*	roots	[17]
		*Thespesia populnea*	barks	[13]
**10**	(−)-fisetinidol	*Bauhinia pentandra*	stems	[23]
		*Rhoicissus tridentata*	roots	[17]
		*Acacia molissima*	woods	[24,25]
**11**	(−)-robinetinidol	*Lysidice rhodostegia*	barks	[26]
**12**	(−)-epigallocatechin	*Thespesia populnea* *Zizyphus jujuba* Mill. var. *spinosa*	barks	[13]
			roots	[11]
**13**	(−)-catechin	*Astilbe rivularis*;	rhizomes	[16]
		*Hovenia dulcis*	stem barks	[10]
		*Juniperus*	stems and	[14]
		*communis var. depressa*	leaves	
		*Plicosepalus curviflorus*	leaves	[27]
		*Euonymus laxiflorus*	trunk bark	[18]
		*Artocarpus xanthocarpus*	leaves	[28]
		*Saraca asoca*		
			roots	[21]
**14**	(*2S*,*3R*)-3,3′,4′,5,7-pentahydroxyflavane-5-*O*-gallate	*Plicosepalus curviflorus*	leaves	[26]
**15**	(−)-afzelechin	*Astilbe rivularis*	rhizomes	[16]
**16**	(−)-gallocatechin	*Euonymus laxiflorus*	trunk barks	[18]
		Champ		
		*Livistona sinensis*	leaves	[12]
**17**	(*2S*,*3R*)-3,3′,4′,5,5′,7-hexahydroxyflavane-4′,5-di-*O*-gallate	*Plicosepalus curviflorus*	leaves	[26]
**18**	(−)-epicatechin	*Wisteria floribunda*	fruit peels leaves leaves roots roots stems and leaves fruits roots barks fruits	[8]
		*Cassia grandis*		[29]
		*Acacia catechu*		[9]
		*Rhoicissus tridentata*		[17]
		*Lysidice rhodostegia*		[25]
		*Juniperus*		[14]
		*communis var. depressa*		
		*Livistona chinensis*		[12]
		*Cassia sieberiana*		[30]
		*Thespesia populnea*		[13]
		*Chaenomeles speciosa*		[19]
**19**	3′-*O*-methylepicatechin-7-*O*-β-d-glucopyranoside	*Juniperus communis* var.*depressa*	stems and leaves	[14]
**20**	3′-*O*-deoxycatechin-3-*O*-α-l-rhamnopyranoside	*Turpinia formosana*	roots	[31]
**21**	(−)-epicatechin-3-*O*-gallate	*Camellia sinensis*	leaves	[9]
		*Rhoicissus tridentata*	roots	[17]
		*Lysidice rhodostegia*	roots	[25]
**22**	(−)-epicatechin-3-*O*-β-d-allopyranoside	*Turpinia formosana*	roots	[31]
**23**	5-*O*-methylepicatechin-7-*O*-β-d-glucopyranoside	*Juniperus communis* var.*depressa*	stems and leaves	[14]
**24**	3,7,4′-trihydroxy-5,3′-dimethoxyflavan-7-*O*-β-d-glucopyranoside	*Styrax suberifolius*	barks	[32]
**25**	(−)-epiafzelechin	*Wisteria floribunda*	fruit peels	[8]
		*Astilbe rivularis*	rhizomes	[16]
		*Cassia grandis*	leaves	[29]
		*Zizyphus jujuba* Mill.	roots	[11]
		*var. spinosa*		
		*Thespesia populnea*	barks	[13]
		*Saraca asoca*	barks	[21]
**26**	(+)-epiafzelechin 7-*O*-β-d-gluco-pyranoside	*Wisteria floribunda*	fruit peels	[8]
		*Saraca asoca*	barks	[21]
**27**	3′-*O*-methylepicatechin-7-*O*-β-d-glucopyranoside	*Juniperus* *communis* var.*depressa*	stems	[14]
**28**	(2*R*,3*R*)-3,5,7,3′,5′-penthahydroxyflavane	*Camellia sinensis*	leaves	[13]
**29**	(*2R*,*3R*)-3,5,6,7,8,3′,5′-heptahydroxyflavane	*Livistona chinensis*	fruits	[12]
**30**	(*2R*,*3R*)-3,5,6,7,8,4′-hexahydroxyflavane	*Livistona chinensis*	fruits	[12]
**31**	(−)-epigallocatechin	*Rhoicissus tridentata*	roots	[17]
		*Camellia sinensis*	leaves	[9]
		*Zizyphus jujuba* Mill.	roots	[11]
		var. *spinosa*		
		*Thespesia populnea*	barks	[13]
**32**	(−)-epigallocatechin gallate	*Camellia sinensis.*	leaves	[33]
**33**	(*2S*,*3S*)-3,5,7,3′,5′-pentahydroxyflavane	*Livistona. chinensis*	fruits	[12]
**34**	dulcisflavan	*Garcinia dulcis*	fruits	[34]
**35**	(−)-6-(5′’*S*)-2-pyrrolidinone-epiafzelechin	*Actinidia arguta*	roots	[35]
**36**	(−)-6-(5′’*R*)-*N*-ethyl-2-pyrrolidinone-epicatechin-3-*O*-gallate	*Camellia sinensis* var. *assamica*	leaves	[36]
**37**	(−)-6-(5′’*S*)-*N*-ethyl-2-pyrrolidinone-epicatechin-3-*O*-gallate	*Camellia sinensis* var. *assamica*	leaves	[36]
**38**	(−)-8-(5′’*R*)-(2-pyrrolidinone-5-yl)-(−)-epicatechin	*Camellia sinensis var. assamica*	leaves	[36]
		*Actinidia arguta*	roots	[36]
**39**	(−)-8-(5′’*S*)-(2-pyrrolidinone-5-yl)-(−)-epicatechin	*Camellia sinensis var. assamica*	leaves	[36]
		*Actinidia arguta*	roots	[35]
**40**	(−)-8-(5′’*R*)-*N*-ethyl-2-pyrrolidinone-epicatechin	*Camellia sinensis* var. *assamica*	leaves	[36]
**41**	(−)-8-(5′’*R*)-*N*-ethyl-2-pyrrolidinone-epicatechin	*Camellia sinensis* var. *assamica*	leaves	[36]
**42**	(−)-8-(5′’*R*)-*N*-ethyl-2-pyrrolidinone-epicatechin-3-*O*-gallate	*Camellia sinensis* var. *assamica*	leaves	[36]
**43**	(−)-8-(5′’*S*)-*N*-ethyl-2-pyrrolidinone-epicatechin-3-*O*-gallate	*Camellia sinensis* var. *assamica*	leaves	[36]
**44**	kopsirachin	*Kopsia dasyrachis*	leaves	[37]
**45**	(−)-uncariagambiriine	*Uncaria gambir*	leaves	[38]
**46**	lotthanongine	*Trigonostemon reidioides*;	leaves	[39]
		*Baliospermum reidioides*	leaves	[39]
**47**	8′-ethylpyrrolidinonyltheasinensin A	*Camellia sinensis*	leaves	[40]
**48**	(−)-epicatechin-(2β→*O*-7; 4β→8)-(+)-catechin	*Camellia sinensis*	leaves	[41]
**49**	(−)-epicatechin-(2β→*O*-7; 4β→8)-(−)-catechin	*Theobroma cacao*	fruits	[42]
**50**	3T-*O*-α-l-arabinopyranosyl-ent-epicatechin-(2α→7,4α→8)-epicatechin	*Theobroma cacao*	fruits	[42]
**51**	3T-*O*-β-d-galactopyranosyl-ent-epicatechin-(2α→7,4α→8)-epicatechin	*Theobroma cacao*	fruits	[42]
**52**	3T-*O*-α-l-arabinopyranosyl-ent-epicatechin-(2α→7,4α→8)-epicatechin	*Theobroma cacao*	fruits	[42]
**53**	procyanidin B4	*Rhoicissus tridentata*	roots	[17]
		*Picea abies*	root barks	[20]
		*Camellia sinensis*	leaves	[41]
**54**	procyanidin B3	*Rhoicissus tridentata*	roots	[17]
		*Picea abies*	root barks	[21]
**55**	fisetinidol-(4β→8)-catechin	*Rhoicissus tridentata*	roots	[17]
**56**	fisetinidol-(4α→8)	*Rhoicissus tridentata*	roots	[17]
**57**	procyanidin B2	*Theobroma cacao*	fruits	[42]
**58**	procyanidin B5	*Theobroma cacao*	fruits	[42]
**59**	methylene 8,8-bis-(+)-catechin	*Potentilla fruticosa*	leaves	[43]
**60**	methylene-6,8-bis-(+)-catechin	*Potentilla fruticosa*	leaves	[43]
**61**	methylene 6,8-bis-(7-*O*-glucosyl)-(+)-catechin	*Potentilla fruticosa*	leaves	[43]
**62**	theaflavin	*Camellia sinensis*	leaves	[41]
**63**	theaflavin-3-galloyl	*Camellia sinensis*	leaves	[41]
**64**	theaflavin-3,3′-galloyl	*Camellia sinensis*	leaves	[41]
**65**	theaflavin-3′-galloyl	*Camellia sinensis*	leaves	[41]
**66**	procyanidin C1	*Theobroma cacao*	fruits	[41]
**67**	procyanidin-C1-tri-gallate	*Theobroma cacao*	fruits	[42]
**68**		*Theobroma cacao*	fruits	[42]
**69**		*Theobroma cacao*	fruits	[42]
**70**	3T-*O*-α-l-arabinopyranosylcinnamtannin B1	*Theobroma cacao*	fruits	[42]
**71**	3T-*O*-β-d-galactopyranosylcinnamtannin B1	*Theobroma cacao*	fruits	[42]
**72**	cinnamtannin B1	*Theobroma cacao*	fruits	[42]
**73**	cinnamtannin A2	*Theobroma cacao*	fruits	[42]
**74**	angular tannin	*Acacia mangium*	barks	[44]
**75**	linear tannin	*Acacia mangium*	barks	[44]
**76**	twice-angular tannin	*Acacia mangium*	barks	[44]

‘‘T’’ and ‘‘ent’’ mean the top unit of the trimeric flavan-3-ols and enantiomer, respectively.

**Table 2 molecules-27-00719-t002:** The numbers, plant sources and corresponding organs of flavan-4-ols.

No.	Name	Plant source	Plant Organ	Ref.
**77**	(*2R*,*4S*)-7-hydroxy-flavan-4-ol	*Morus alba*	leaves	[49]
**78**	(*2R*,*4S*)-2′,4′-dihydroxy-2*H*-furan-(3′’,4′’:8,7)-flavan-4-ol	*Morus alba*	leaves	[49]
**79**	(*2S*,*4R*)-7,4′-dihydroxy-6,8-dimethyl-4,2′’-oxidoflavan-5-*O*-β-d-glucopyranoside	*Abacopteris penangiana or* *Pronephrium penangiana*	rhizomes	[50]
**80**	eruberin A	*Abacopteris penangiana or Pronephrium penangiana*	rhizomes	[50]
**81**	(*2S*,*4R*)-6,8-dimethyl-4′-methoxy-4,2′’-oxidoflavan-5,7-di-*O*-β-d-glucopyranoside	*Abacopteris penangiana or Pronephrium penangiana*	rhizomes	[50]
**82**	abacopterin G	*Abacopteris penangiana or Pronephrium penangiana*	rhizomes	[50]
**83**	(*2S*,*4R*)-7,4′-dihydroxy-6-methoxymethyl-8-methyl-4,2′’-oxidoflavan-5-*O*-β-d-glucopyranoside	*Abacopteris penangiana or Pronephrium penangiana*	rhizomes	[50]
**84**	(*2S*,*4R*)-5,7-dihydroxy-4,4′-dimethoxy-6,8-dimethylflavan-5-*O*-β-d-6-acetylglucopyranoside-7-*O*-β-d-glucopyranoside	*Abacopteris penangiana or Pronephrium penangiana*	rhizomes	[50]
**85**	(2*S*,4*R*)-4,5,7-trihydroxy-4′-methoxy-6,8-dimethylflavan-5-*O*-β-d-6-glucopyranoside-7-*O*-β-d-glucopyranoside	*Abacopteris penangiana or Pronephrium penangiana*	rhizomes	[50]
**86**	6′’-*O*-acetyltriphyllin A	*Abacopteris penangiana or Pronephrium penangiana*	rhizomes	[51]
**87**	eruberin B	*Abacopteris penangiana or Pronephrium penangiana*	rhizomes	[50,52]
**88**	4′-hydroxypneumatopterin B	*Abacopteris penangiana or Pronephrium penangiana*	rhizomes	[51]
**89**	(*2S*,*4R*)-4,5,7-trihydroxy-4′-methoxy-6,8-dimethylflavan-5-*O*-β-d-6-acetylglucopyranoside-7-*O*-β-d-glucopyranosid	*Abacopteris penangiana or Pronephrium penangiana*	rhizomes	[50]
**90**	triphyllin A	*Abacopteris penangiana or Pronephrium penangiana*	rhizomes	[53]
**91**	eruberin C	*Abacopteris penangiana or Pronephrium penangiana*	rhizomes	[52]
**92**	abacopterin D	*Abacopteris penangiana or Pronephrium penangiana*	rhizomes	[53]
**93**	abacopterin K	*Abacopteris penangiana or Pronephrium penangiana*	rhizomes	[54]
**94**	(*2R*,*4S*)-6,8-dimethyl-7-hydroxy-4′-methoxy-4,2′’-oxidoflavan-5-*O*-β-d-glucopyranoside	*Abacopteris penangiana or Pronephrium penangiana*	rhizomes	[55]
**95**	(*2R*,*4S*)-6,8-dimethyl-7-hydroxy-4′-methoxy-4,2′’-oxidoflavan-5-*O*-β-d-6′’-*O*-acetyl-glucopyranoside	*Abacopteris penangiana or Pronephrium penangiana*	rhizomes	[56]
**96**	(*2S*,*4R*)-7-hydroxy-6-methoxymethyl-8-methyl-4,2′’-oxidoflavan-5-*O*-β-d-glucopyranoside	*Abacopteris penangiana or Pronephrium penangiana*	rhizomes	[56]
**97**	abacopterin E	*Abacopteris penangiana or Pronephrium penangiana*	rhizomes	[57]
**98**	abacopterin C	*Abacopteris penangiana or Pronephrium penangiana*	rhizomes	[56]
**99**	abacopterin A	*Abacopteris penangiana or Pronephrium penangiana*	rhizomes	[56]
**100**	abacopterin B	*Abacopteris penangiana or Pronephrium penangiana*	rhizomes	[56]

**Table 3 molecules-27-00719-t003:** The numbers, plant sources, and corresponding organs of isoflavan-4-ols.

No.	Name	Plant Source	Plant Organ	Ref.
**101**	maackiain	*Astragalus cicer*;	roots	[58,59]
		*Astragalus membranaceus*;	roots	
		*Astragalus mongholicus*;		
		*Astragalus trojanus*	roots	
**102**	(6a*R*,11a*R*)-10-hydroxy-3,9,10-dimeth oxypterocarpan	*Astragalus membranaceus*;	roots	[60,61]
**103**	(6a*R*,11a*R*)-3,9,10-trimethoxypterocarpan	*Astragalus membranaceus*;	roots	[61]
		*Astragalus mongholicus*	roots	
**104**	vesticarpan	*Astragalus membranaceus*	roots	[62]
**105**	(−)-methylinissolin	*Astragalus membranaceus*	roots	[62]
**106**	(−)-methylinissolin-3-*O*-β-d-{6′-[2′’-(*E*)-ene-penyl]}-glucoside	*Astragalus membranaceus*	roots	[62]
**107**	(−)-methylinissolin-3-*O*-β-d-(6′-acetyl)-glucoside	*Astragalus membranaceus*	roots	[62]
**108**	(−)-methylinissolin-3-*O*-β-d-glucoside	*Astragalus membranaceus*	roots	[62]
**109**	licoagroside D	*Astragalus membranaceus*	roots	[62]

**Table 4 molecules-27-00719-t004:** The numbers, plant sources and corresponding organs of flavan-3,4-ols.

No.	Name	Plant Source	Plant Organ	Ref.
**110**	guibourtacacidin	*Acacia farnesiana*	heartwoods	[63]
**111**	mollisacacidin	*Acacia farnesiana*	heartwoods	[63]
**112**	leucopelargonidin	*Saraca asoca*	barks	[25]
**113**	leucocyanidin	*Saraca asoca*	barks	[25]
**114**	teracacidin	*Acacia galpinii*	heartwoods	[63]
**115**	melacacidin	*Acacia nilotica* subsp. indica	heartwoods	[63]
**116**	(+)-2,3-*trans*-3,4-*cis*-peltogynol	*Acacia melanoxylon*	barks	[64] [65]
**117**	(−)-2,3-*cis*-3,4-*cis*-peltogynol	*Acacia peuce*;	barks	[64] [65]
		*Acacia carneorum*;	barks	
		*Acacia crombiei*;	barks	
		*Acacia fasciculifera*	barks	
**118**	[4-*O*-4]-bis-(2,3-*cis*-3,4-*trans*-3,3′,4′,7,8)-pentahydroxyflavan	*Acacia peuce*;	barks	[66]
		*Acacia carneorum*;	barks	
		*Acacia crombiei*;	barks	
		*Acacia fasciculifer*	barks	
**119**	2,3-*cis*-3,4-*trans*-3,3′,4′,7,8-pentahydroxflavan-[4-*O*-4]-2,3-*cis*-3,4-*cis*-3,3′,4′,7,8-pentahydroxyflavan	*Acacia peuce*;	barks	[66]
		*Acacia carneorum*;	barks	
		*Acacia crombiei*;	barks	
		*Acacia fasciculifera*	barks	
**120**	[3,4,3′,4]-*O*,*O*-linked-bis-(2,3-*trans*-3,4-*cis*-3′,4′,7-trihydroxyflavan)	*Acacia peuce*;	barks	[67]
		*Acacia carneorum*;	barks	
		*Acacia crombiei*;	barks	
		*Acacia fasciculifera*	barks	
**121**	2,3-*trans*-3,4-*trans*-2′,3′-*trans*-3′,4′-*cis*-diastereoisomer	*Acacia peuce*;	barks	[67]
		*Acacia carneorum*;	barks	
		*Acacia crombiei*;	barks	
		*Acacia fasciculifera*	barks	

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
