# Peer review of "Flavanols from Nature: A Phytochemistry and Biological Activity Review"

_molecules, 2022, doi:10.3390/molecules27030719_

Round 1
Reviewer 1 Report
Thank you for revising in accordance to my suggestions.
This is a meaningful and helpful review that provides information on the origin, structure, and bioactivity of flavonols.
Author Response
Journal: Molecules
Manuscript Number: molecules-1506535
Title: “Flavanols from Nature: Phytochemistry, Biological Activity Review”
Author(s): Yu Luo †, Yuqing Jian †,*, Yingkai Liu, Sai Jiang, Muhammad Daniyal and Wei Wang*
For the “Reviewer 1”
- This is a meaningful and helpful review that provides information on the origin, structure, and bioactivity of flavanols.
Answer:
Special thanks to you for your comments and suggestions.
Reviewer 2 Report
The manuscript topic related to the phytochemistry and biological activity of flavanols is of interest and adequate for this Journal. However, in this stage, the review is not balanced and organized enough and would need restructuring in order to be recommended for publication. In general, the novel perspective and critical view are lacking.
Author Response
- In this stage, the review is not balanced and organized enough and would need restructuring in order to be recommended for publication.
Answer:
We are very grateful to your comments and thoughtful suggestions.We have divided the table 1 into four different tables (tables 1-4) and deleted Figs. 2 and 3, optimized the content of Fig. 2 (Fig. 2. of the revised manuscript is the Fig. 3. of the first manuscript), hope it can make the expression of the revised manuscript clearer. Moreover, a large number of modifications and optimizations have been made in the revised manuscript, and the language expression has also been further polished.
Reviewer 3 Report
This manuscript reviews the available literature on the importance of natural flavanols and their biological activity as promising structures that can provide evidence for the development of new drugs. Here, the authors focus on the 121 flavanols and their derivatives which are subdivided into four categories: flavan-3-ols, flavan-4-ols, isoflavan-4-ols, and flavan-3,4-ols.
The manuscript is clearly described, well motivated and pertinent to the journal. The plan structure and tasks are clearly presented.
Also, I find the described biological activity to be balanced and the relevant literature has been cited.
My comments are:
- Fig. 2 does not give a clear idea of the classification of flavanols and it might be better to remove
- Fig. 3 The Authors must provide a good figure quality. It is not clear what the numbers in thebracketsmean as well as different colors. A detailed description must be given. This would strongly support the findings presented and would allow the readers a better interpretation of the figure.
- Fig. 4 can be removed because the presence of each of the 121 flavanols is presented in the different plant species in Table 1. I propose that the authors add the percentages of flavanols in the different species as an additional column in Table 1.
- Line 215 The Authors must provide the correct citation …….”In previous studies, (-)-epicatechin is an important catalyst of enzymatic lipid peroxidation of biomembranes and plasma lipoproteins” …..
I find this review a well written, clear and an interesting, timely report. So my recommendation would be to accept after a minor revision.

Reviewer 4 Report
The authors have presented a review on flavanols, paying particular attention to chemistry and biological activity. Overall, the topic is interesting and the review is comprehensive, informative and potentially of interest to a wide range of audience. However, following suggestions are recommended:
-there are several typewriting errors and some sentences are rambling, please modify (i.e. page 2 line 46-46 “in 54 species 29 families” please change with “in 54 species and 29 families”).
-when authors start to describe the polymerization of flavan-3-ols they have to highlight that they are not still the same classes of compounds but they start to analyze proanthocyanidins
-The quality of the Figure 3 and 4 needs to be improved. It is hard to read along the axis and also the corresponding title. In my ipinion thae must be divided in different pages, not in the same page.
-The table one is too big. Please divide into three, four different table selecting a criteria base on the classification for example(flavan-3-ols, flavan-4-ols, isoflavan-4-ols, 17 and flavan-3,4-ols)
Author Response
Journal: Molecules
Manuscript Number: molecules-1506535
Title: “Flavanols from Nature: Phytochemistry, Biological Activity Review”
Author(s): Yu Luo †, Yuqing Jian †,*, Yingkai Liu, Sai Jiang, Muhammad Daniyal and Wei Wang*
For the “Reviewer 4”
- There are several typewriting errors and some sentences are rambling, please modify (i.e. page 2 line 46-46 “in 54 species 29 families” please change with “in 54 species and 29 families”).
Answer:
Special thanks to you for your comments and suggestions. We have revised the several typewriting errors and some rambling sentences. Page 2 line 46-46 “in 54 species 29 families” have been changed by “in 52 species and 29 families”. (in line 44).
- When authors start to describe the polymerization of flavan-3-ols they have to highlight that they are not still the same classes of compounds but they start to analyze proanthocyanidins.
Answer:
Thank you for your comments and suggestions. Proanthocyanidins are condensation of flavan-3-ols. To prevent readers from being confused, we changed proanthocyanidins by oligomeric flavan-3-ols.
- The quality of the Figure 3 and 4 needs to be improved. It is hard to read along theaxis and also the corresponding title. In my ipinion thae must be divided in different pages, not in the same page.
Answer:
Thank you for your comments and suggestions. We have improved the quality of Fig. 3 (which named Fig. 2 in the revised manuscript) and give a detailed description such as Families &Species and the corresponding quantity of compounds. Moreover, mainly 11 species plants rich-in flavanols are marked in blue color. We have deleted Fig. 4.( Numbers and percentages of flavanols in different species.) and Fig. 2 (The classification of flavanols and their quantity in each category.) in the the first manuscript.
- The table one is too big. Please divide into three, four different table selecting a criteria base on the classification for example (flavan-3-ols, flavan-4-ols, isoflavan-4-ols, and flavan-3,4-ols).
Answer:
Thanks for your constructive comments. We have divided the table 1 into four different tables (tables 1-4) in the revised manuscript.
Round 2
Reviewer 2 Report
The manuscript topic related to the phytochemistry and biological activity of flavanols is of interest and adequate for this Journal. In the revised version of the manuscript, the organization and structure of the work are much improved, giving more clear insight into the content. In this form, I would recommend the publication, taking into account that in the title word review should be properly spelled, without cutting it into 2 rows (re-view).